# Indoor WiFi-Beacon Dataset Construction Using Autonomous Low-Cost Robot for 3D Location Estimation

**Suleiman Abu Kharmeh** [1,*] **, Emad Natsheh** [1] **, Batoul Sulaiman** [2] **, Mohammad Abuabiah** [3] **and Saed Tarapiah** [4]

1   Department of Computer Engineering, An-Najah National University, Nablus P.O. Box 7, Palestine; e.natsheh@najah.edu
2   Department of Computerized Mathematics, An-Najah National University, Nablus P.O. Box 7, Palestine; batoolsulaiman11@gmail.com
3   Department of Mechatronics Engineering, An-Najah National University, Nablus P.O. Box 7, Palestine; m.abuabiah@najah.edu
4   Department of Telecommunication Engineering, An-Najah National University, Nablus P.O. Box 7, Palestine; s.tarapiah@najah.edu
*   Correspondence: sabukharmeh@najah.edu

**Abstract:** Datasets used for artificial-neural-network and machine-learning applications play a vital role in the research and application of such techniques in solving real-life problems. The construction and availability of large datasets to be used in the off-line phase of ANN training is usually a crucial and time-consuming step towards system construction. In this work, a framework for autonomous construction of a diverse, extensive, and open dataset* with built-in redundancy is demonstrated. As part of the framework, a low-cost robot using off-the-shelf components is built that constructs the dataset autonomously. The robot includes a controller network with multiple WiFi-transceiver nodes for collecting received-signal-strength indicators (RSSIs) at various elevation points throughout the building. All nodes are configured with direct internet access to streamline the data collection towards an online database that is constructed as part of this framework. Preliminary validation and analysis of the dataset are discussed, and an exploration of the application domain of the dataset is carried out. Moreover, this paper investigates the effect of the height of the hand-held mobile WiFi antenna attached to the robot on the received power strength of the WiFi signal.

**Keywords:** indoor localization; dataset construction; low-cost robot; artificial neural network; signal-strength indicator

## 1. Introduction

The indoor localization process has increased in popularity and received significant attention from both academia and industry. However, even with the huge existing number of studies on this topic, localization is still considered a big challenge and there is no optimal solution for all existing applications. Nowadays, various indoor networks, such as WiFi, Bluetooth, Zigbee, and more, have been used to obtain received-signal-strength (RSS) values to estimate an object's position, using it as the primary feature. Unlike other wireless technologies, the WiFi-fingerprint-localization system [1] does not require the deployment of extra infrastructure, as WiFi routers are already widely distributed in most indoor environments, such as residential buildings, commercial shops, airports . . . etc. Therefore, WiFi-based IPS has become one of the most promising systems these days.

The WiFi-fingerprinting method is composed of two main phases: the training phase, for fingerprint-radio-map generation, and the estimation phase, for the user-location estimation process. The radio-map generation process is time-consuming and expensive, which is considered a drawback of this method. Therefore, the traditional systems for the radio-map creation process, which are based on manual collection such that the surveyor

needs to walk through the whole localization area to collect the WiFi fingerprints, are labor-intensive and time-consuming systems, especially in large-scale areas. As a result, a contemporary solution for the WiFi-fingerprinting training phase is the automated method (robot-based method) [2,3], where a robot collects RSS and builds a radio map to be able to perform the location-estimation process.

Indoor robot-based solutions have many potential applications in daily life, such as home care, data collection, object finding, and emergency support [4,5]. On the other hand, designing autonomous robots is considered a critical issue since it requires the continuous computation of various characteristics of the robot, such as width, height, position of the sensors mounted on the robot, or wheel diameter, which can be used to make assumptions about the movement of the robot [6].

A robot developed by researchers in [7] is capable of generating a map using LiDAR sensors and simultaneous localization and mapping (SLAM) methods. The robot was equipped with an RGBD camera to improve the efficiency of the WiFi radio map, especially in featureless areas such as corridors. Despite this, the developed robot scanned 3000 points in 2 h without providing information about the experimental environment and the mean error of the positioning system.

A robot's power needs become crucial when operating in large indoor environments. Researchers used a robot equipped with LiDAR and SLAM to construct a dual WiFi radio map (2.4 and 5.0 GHz) in [8] and, using a Bayesian probabilistic model and K-nearest-neighbor algorithm, they were able to reach a mean error of 2.4 m. The robot in this experiment consumed between 60 and 82 Wh and was able to scan two university floors in 94 min, making it an extremely power-efficient device.

According to [9], a robot-based method was adopted to collect data more effectively and quickly. By analyzing the map's reliability and collecting additional learning data, a reliable WiFi radio map (WRM) was constructed using the SLAM-based data-collection-and-analysis (SDCA) technique. The WRM construction was improved by 12.9% with this method. Furthermore, they developed a robust tracking algorithm that can handle fluctuations in WiFi signals after collecting the data called the extended-Viterbi-algorithm and signal-fluctuation-matrix fusion tracking method (EVSFM). However, the author used a small environment with too many access points to achieve high-quality results, where the mean positioning error was about 1 m. In most indoor environments, this is not possible.

In large indoor environments, RSS radio maps can be time-consuming to produce. Researchers from [10] solved this problem by using a robot to collect RSS at various locations in the environment. They also used interpolation methods to estimate RSS values at unmeasured locations. In addition, they applied machine-learning algorithms during the online phase to estimate the object's current position and obtained a 2.21 m error. Furthermore, no map of the environment was created, and the researchers did not provide valuable information about the environment. Only 10 reference points were used in the offline stage of the study.

Indoor positioning reduces the time required to scan the environment, as well as the possibility of dealing with changes in the environment over time. To keep Bluetooth RSS maps up to date, the researchers of [11] developed a robot that can update the fingerprint database frequently. The method was tested on 210 m$^2$, and the mean error was 1.84 m. Within 27 min, the robot scanned the entire environment and generated a route between the specified reference points. According to [12], the researchers used the Gaussian Process Regression-conditioned Least Square Generative Adversarial Network (GPR-GAN) method to solve constrained spaces in the environment. Despite the authors' inability to handle the issue of reaching a small part of the environment with the robot, they reduced the mean error distance to 1.98 m over the environment.

Previous studies have shown that the role of robots is important in improving the efficiency of data collection, and this has an important role in improving the efficiency of the model. Therefore, this study focuses on how the strength of the received WiFi signal

is affected by the height of the handheld WiFi antenna, which is connected to the robot. Different sets of antenna heights are considered.

In the rest of the paper, the related work is discussed first. The existing literature and research relevant to using Wi-Fi beacons in localization is explored. Then, an overview of the overall system architecture is presented, outlining its key components, and functionalities. A detailed description of the robot sub-system follows, where an overview of the robot's external structure, physical design, and characteristics is discussed. Additionally, a schematic description of the robot's electronic system is provided to offer a visual representation of its internal layout and connections. The next section details the robot's operation and its sub-modules. Then, the paper discusses the indoor-environment infrastructure, covering the setup necessary to support the robot's operations in an indoor setting. The backend database and web-server interface is then explored, outlining the underlying technology and functionality enabling interaction with the robot. Then, the paper presents the results obtained, along with a comprehensive discussion analyzing the implications and significance of the results. A conclusion section summarizes the main points discussed in the paper and provides a final reflection. Finally, a list of references is included to acknowledge the sources cited throughout the paper, ensuring academic integrity and facilitating further research.

## 2. Related Work

There are indoor-positioning methods other than indoor networks. Based on the intensities and luminaries of indoor light, some researchers extended the boundaries of indoor positioning. In [13], an experimental robot captured frequency information from four LED transmitters with different frequencies. The robot is equipped with a LiDAR sensor that maps the room and two cameras capable of measuring light intensity and frequency. Although the experimental results reached a mean error of 80 cm, this method appeared to be unsuitable for localization in large environments. In [14], the researchers used light luminaries to predict position. A LiDAR-based SLAM map was generated by a robot that reads the visible light and stores it with a real-time position estimated using the HTC Vive. To predict the robot's current location, they used KNN, support-vector regression (SVR), and decision trees. The robot used in the research was a vacuum robot with an added photodiode to measure light intensities, and they achieved great results in terms of positioning error, which was between 8.4 and 14.4 cm. However, the experiment was extremely expensive, as the HTC Vive tracking system was used only in one room, which had a 43.4 $m^2$ area, and the cost would be high if applied to the entire building floor. In addition, it is only used for positioning trained robots.

In terms of indoor positioning, WiFi is not the only network that caught researchers' attention. The researchers in [3] developed a ZigBee network radio map and a nonlinear-regression model to locate the robot's location, which is good for dynamic environments due to its continuous database updates. In a 145 $m^2$ environment, the robot visited 531 reference points with a mean error of 0.81. Although the robot can be used in any dynamic environment, it cannot avoid moving obstacles. Other researchers chose Bluetooth RSS as fingerprints in indoor positioning, and in [15] LiDAR and a camera were used as wearable devices to scan the environment and create SLAM-based maps. The camera is responsible for labelling the rooms in the environment, and the helmet records Bluetooth signals to be used to estimate position in the future. Other researchers suggested RFID tags as a solution. A study in [16] achieved a 1.22 m error rate for RSS-data collection over a 45 $m^2$ environment by using RFID tags and two-wheel robots. Table 1 summarizes a set of previous studies in this field. Ref. [17] demonstrated the use of Bluetooth low-energy (BLE) beacons for constructing a location-aware infrastructure by using unsupervised learning methods. Data collection, storage, and post-processing techniques in [17] were of particular interest. Paper [18] suggested using support-vector regression (SVR) for mobile-target localization in indoor environments. The techniques discussed in [18] could be applied to the dataset collected in this paper and would lead to an interesting future study.

**Table 1.** Recent studies in the field of indoor positioning.

| Ref. | Year | Robot Type | Environment Area | Map Acquisition | Location Estimation | Positioning Method | Time | Observations | Localization Error | Scope |
|------|------|-----------|-----------------|----------------|---------------------|-------------------|------|--------------|-------------------|-------|
| [3] | 2019 | Custom | 145 m$^2$ | SLAM | ASMF | ZigBee | NA | 531 | 0.81 m | Generic (ZigBee) |
| [8] | 2020 | Industrial (Pioneer 3-DX) | 2 university floors | SLAM | KNN Bayes with Gaussian process | WiFi | 94 min | NA | 2.4 m | Generic (smartphones) |
| [9] | 2020 | Industrial (Kobuki) | 89 m$^2$ | SLAM | EVSFM | WiFi | 2.5 h | 825 | 1 m | Generic |
| [10] | 2019 | Custom | unknown | Pre-defined | KNN | WiFi | NA | 10 | 2.21 m | Generic |
| [11] | 2018 | Industrial (Pioneer 3-DX) | 210 m$^2$ | SLAM | PDR | Bluetooth | 27 min | NA | 1.84 m | Generic |
| [12] | 2020 | Industrial (Kobuki) | 700 m$^2$ | SLAM | GPR-GAN | WiFi | NA | NA | 1.98 m | Generic |
| [13] | 2021 | Industrial (Kobuki turtlebot) | One room | SLAM | PDR | VLP | NA | NA | 0.8 m | Robot specific |
| [14] | 2021 | Industrial (Roborock s50 vacuum) | One room (43.4 m$^2$) | SLAM | Ridge regression | VLP | NA | 7344 | 84 to 144 mm | Robot specific |
| [15] | 2020 | Custom | One floor | SLAM | SLAM | BLE | NA | NA | NA | Wearable device |
| [16] | 2021 | Industrial (Pioneer 3-DX) | 45 m$^2$ | Pre-defined | Odometry | RFID | NA | NA | 1.22 m | Robot specific |
| [18] | 2022 | Custom | 100 m$^2$ | Pre-defined | Support-vector regression | WiFi | NA | NA | 0.8528 m | Generic (smartphones) |

For an efficient RSS-based IPS, there are various factors that should be considered, such as human-body effects, building material, device diversity, antenna polarization, and sensor-placement height, and a good understanding of RSSI behavior with these factors will lead to an accurate, efficient, and effective IPS [19]. Various studies have discussed these factors, such as [20], in which the author introduces a detailed explanation of the building material and environmental effects on WiFi RSSI measurements, such that their conclusion shows that metal makes a significant contribution to signal-strength fluctuations and that isolators such as wood and plastic contribute to a reduction in signal strength. The author of [21] discussed the effect of device diversity in the localization process, and their results showed significant differences in the behavior of different devices in a practical environment. The authors of [22] demonstrated the use of a handheld device operated manually by a human for a dataset construction that maps the same area of study as this paper. The authors then used the collected dataset for constructing, training, and then testing an AI model used for location estimation. The size of the collected dataset in [22] is not clear, but it is apparent that the number of collected samples was not sufficient for constructing an accurate ANN, hence the use of the BSI method to generate a denser database and attain an effective ANN solution. The authors in [23] considered the polarization of the antenna and its effect on the accuracy of the location-estimation identification. Although the authors of [23] discussed mathematically how the RSSI value is related to the distance between the transmitter and the receiver, their focus was on the polarization of the transceivers and not on the effect of the distance between the transmitter and the receiver on the location estimation, which is the focus of this paper.

Most of the existing studies overlooked the effect of the height of the receiver antenna. Therefore, this paper investigates the effect of the height of the hand-held mobile WiFi antenna attached to the robot on the received power strength of the WiFi signal. There are different heights of antennas being studied that are spaced 60 cm apart. As RSSI is used for indoor navigation, the default height was set to 1.5 m, which is thought to be the average height of a pedestrian's hand-held smartphone.

## 3. Overall System Architecture

A systematic approach was established at an early stage in this project for data collection, storage, and retrieval for the purpose of this work. The overall system architecture is demonstrated in Figure 1. The main component of the system is the autonomous robot, which was constructed to facilitate the autonomous collection of the WiFi beacons in the target area. The autonomous robot is described in Section 3.1.

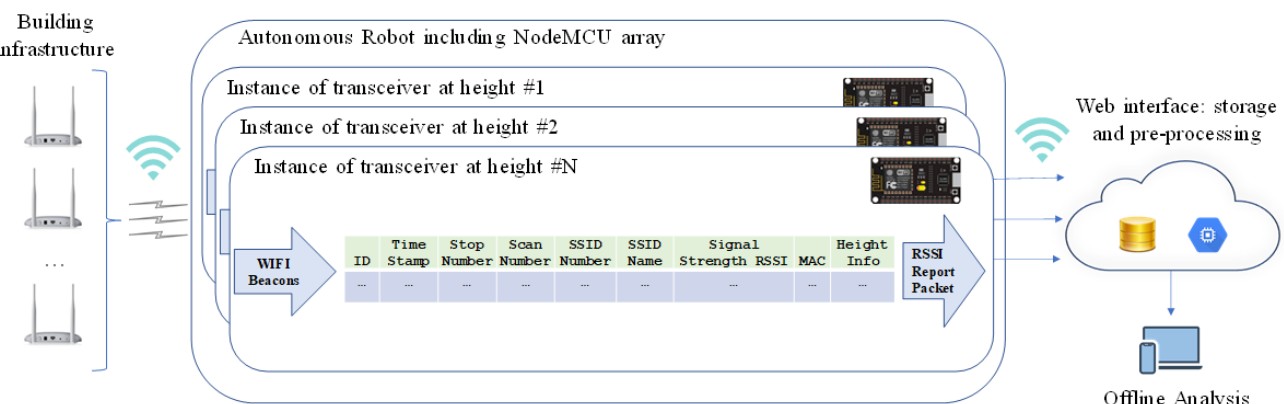

**Figure 1.** Overall system architecture.

The building infrastructure was of high significance in this study. It was necessary for the installed WiFi access points (APs) to support the management frames periodically broadcasted by WiFi APs announcing the presence of the WiFi LAN and its available

parameters. In addition, a track was installed to aid the robot in data collection at predefined locations. The building infrastructure is discussed in Section 3.2.

Finally, a web interface was implemented to aid the robot in data storage due to the large size of the collection, which would always be larger than the internal storage available in the target low-cost microcontroller systems of choice. The web interface and associated components are described in Section 3.3.

### 3.1. Robot Sub-System Description

As part of this work, an autonomous robot was constructed to collect all relevant WiFi-hotspot information from various heights throughout the desired indoor space. This information was collected from the broadcasted WiFi-beacon frames. The collected information includes service set identifiers (SSIDs), received-signal-strength indicator (RSSIs), encryption, and media-access-control (MAC) addresses. The details of the robot are discussed in the remainder of this section.

### 3.1.1. External Overview

An overview of the robot can be seen in Figure 2b. It includes three NodeMCU boards mounted at selected heights on an antenna tower, as seen in Figure 2b. The WiFi-transceiver logic-board (NodeMCU) modules are connected to a dedicated 8 V power-supply battery pack, as seen in Figure 2b: $4 \times 4$ V Li-ion batteries are used, each two in series, to provide the WiFi transceivers with the necessary power during the extended hours of operation (more than four hours of operation for each run of data collection). Four DC motors are each connected to an individual wheel for precise movements and a motor power-supply sub-circuit (L298N), which is connected to a separate power-supply battery pack (8 V input using $2 \times 4$ V Li-ion batteries in series) mounted at the bottom of the robot to ensure stability by having a low center of gravity in addition to ensuring simple cable routing and an optimal board layout. Two separate power supplies are used in order to isolate the mechanical sub-system from the logic sub-systems. This isolates the logic boards from the electrical noise usually produced by mechanical sub-systems. The body of the robot mainly consists of synthetic polymers that impose minimum interference with the Wi-Fi transceivers. In the final revision of the robot, the Wi-Fi transceivers were mounted on a wooden pole for the same reason. In addition, the main control unit and the client transceiver are linked using shielded CAT 5 network cables to reduce interference. Those precautionary measures ensure minimum impact of the robot itself on the measured signal strength and related communications.

Two infrared transmitters and sensing modules for navigation are connected to the front of the robot for navigation, as seen in Figure 2a. The robot uses infrared (IR) transmitters and sensors to sense the track on which is it intended to traverse. To guarantee accurate displacement, many experiments were carried out by investigating different factors, such as the velocity of the wheels, the relative rotation of the wheels, and tracking module calibrations.

### 3.1.2. Schematic Description

The schematic of the overall electronic system is depicted in Figure 3. The main control board also functions as a WiFi transceiver, for which a NodeMCU logic board is used. The number of logic inputs/outputs from the main controller to each client controller is reduced to two digital IOs: RX and TX. The TX connection is shared between the two client boards, which enables the main controller to transmit to both clients in a broadcast manner. The RX connection is not shared, which results in the client boards being able to relay feedback information individually to the main controller. The connections from the main control board towards the motor-control sub-circuit are achieved using two digital outputs: left-drive and right-drive control signals. The motor-drive signals are treated as analog signals using pulse-width modulation (PWD), which is sufficient to drive the robot to the designated signal-collection points. In addition, this setup also gives complete control of

the speed and direction of movement of the robot to the main control board. Finally, each of the two infrared sensors is connected to the main controller board using two digital-input connections, which are used as an indication of the path markings and enable the robot to follow a predefined path throughout the building in addition to identifying specific checkpoints for WiFi-signal-strength collection.

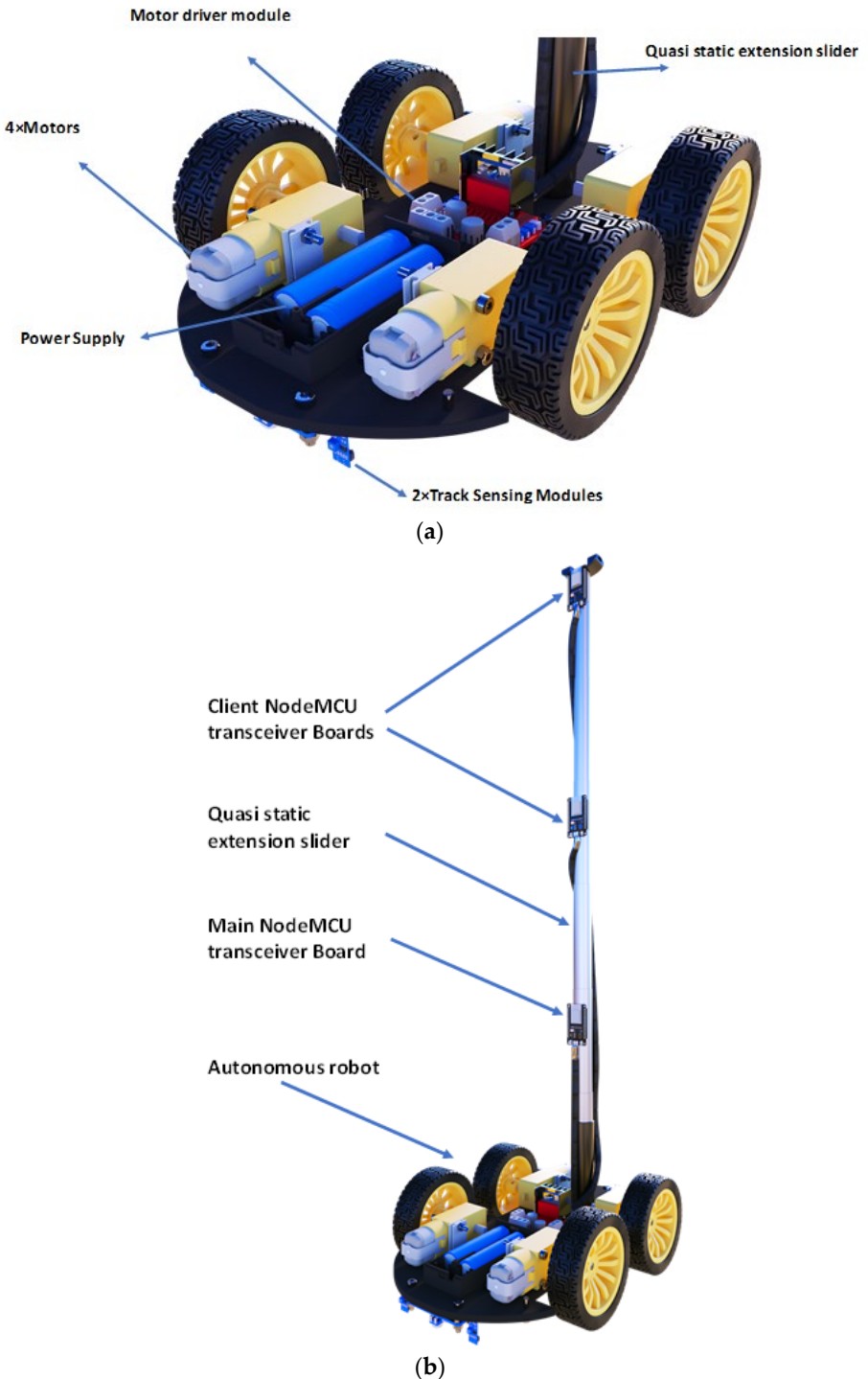

**Figure 2.** (**a**) Robot base unit (top view); (**b**) robot overview (tower height not to scale).

Finally, the power of the mechanical part of the system is provided by two Li-ion batteries connected in series to provide the motors with 8 V of power. The logic and WiFi transceivers are powered using a more powerful setup of four Li-ion batteries, each two

batteries are connected in series, and the two sets are connected in parallel to provide 8 V of power with double the wattage. The two sets of two series-connected batteries are connected in parallel to provide the additional power needed for all WiFi transceivers, which are usually power-hungry. In addition, the system spends a fraction of the time moving between checkpoints and most of the time in WiFi-signal-strength collection. About 3% of the time is spent moving, whereas 97% of the time is spent collecting the required data. This was determined based on many development cycles monitoring power and performance and applying multiple improvements to the system to reach the current minimalist yet very effective system.

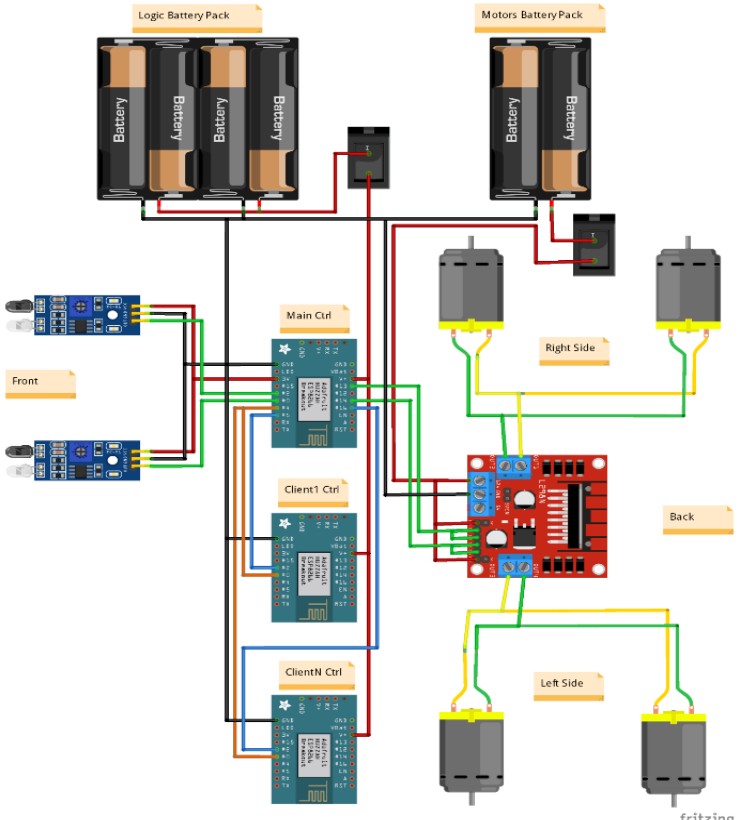

**Figure 3.** Overall robot schematic.

### 3.1.3. Operation

The infrared sensors transmit an infrared beam below the robot and sense the amount of light reflected from the ground beneath the robot. The output of those sensors is an analog signal that depends on the amount of light reflected from the pre-defined track. This analog signal is then converted into a binary signal indicating whether the robot is still on the pre-defined path or has moved in the wrong direction. This binary indicator is then used by the main control unit, which controls the robot's motors, adjusting its movements to the correct path. In addition, the information conveyed by the infrared sensors is used to identify the pre-defined checkpoints at which the robot is intended to pause and collect WiFi-signal-strength information.

### A.     Main Control System

The robot is mainly split into a main control unit and client control units. The flow chart of the main control unit is demonstrated in Figure 4b. Its main function is controlling the speed and direction of the robot, as explained earlier. In addition, it functions as a driver of the client control units. When it detects a checkpoint (clearly marked on the path, as seen in Figure 4a), it sends a trigger to all connected client units instructing them to start collecting the required samples.

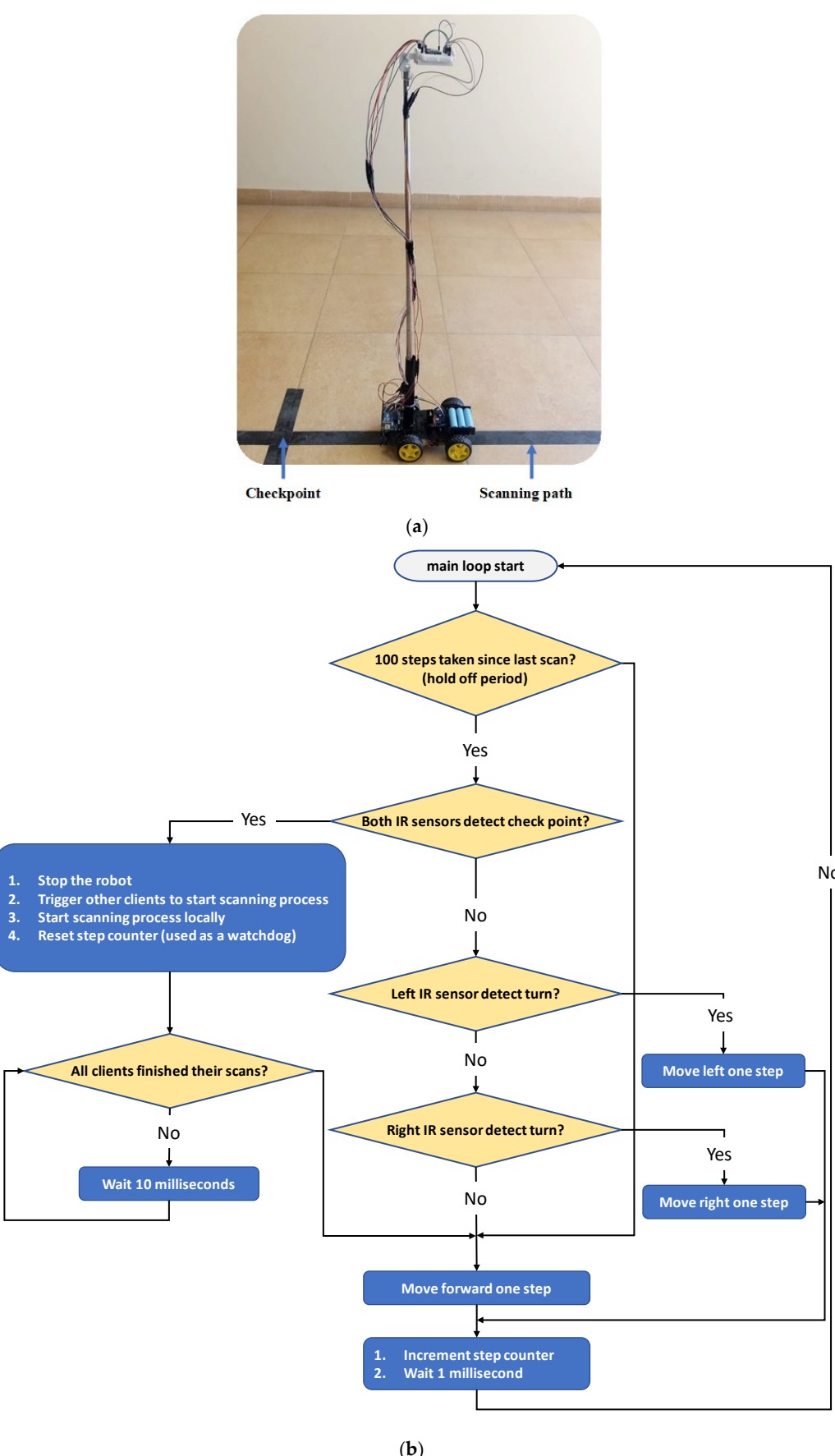

**Figure 4.** (**a**) Earlier version of the robot (for demonstration purposes); (**b**) flowchart of the main control unit.

B. Client WiFi Sub-Systems

The client WiFi-unit control sub-system is demonstrated in Figure 5. Its main task is to collect WiFi samples when instructed to do so by the main controller unit. It attaches a serial number to each sample collected and sends all samples to the remote database over WiFi using HTTP post protocol. The sampling process is performed in a number of prespecified iterations. The effect of height on the received signal strength was of significant importance to this study; hence, the robot was equipped with two additional client sub-systems, one for each height of interest in addition the baseline main controller. At each target point, the measurement data were collected at different height levels, namely, 30 cm, 90 cm, and 150 cm. The main control system was 30 cm high, whereas the client sub-systems were 90 and 150 cm high.

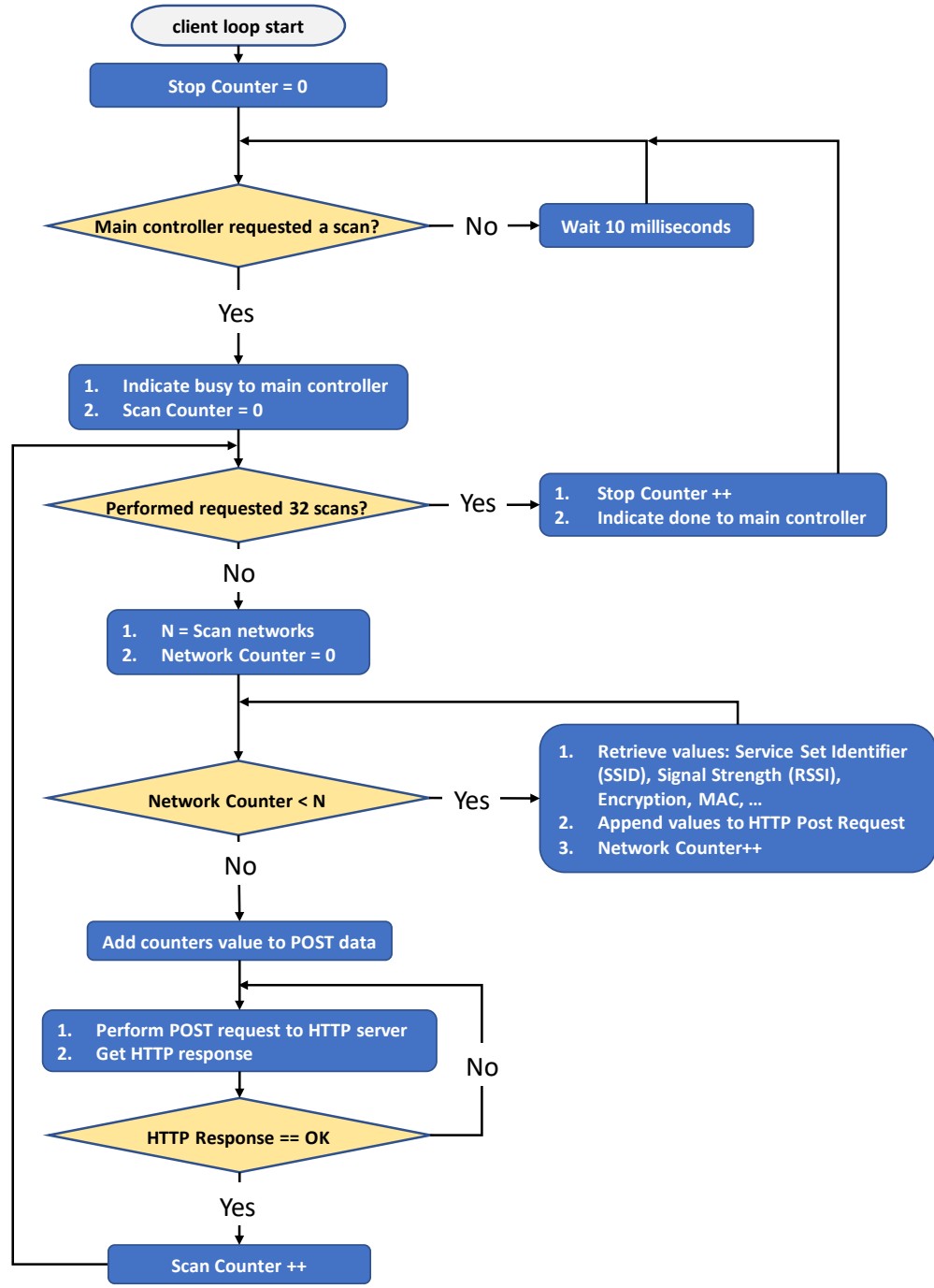

**Figure 5.** Flowchart of the scanning process for client WiFi units.

### 3.2. Indoor-Environment Infrastructure

The indoor environment used for the experiments was the second floor of the engineering college at An-Najah National University, which covers an area of approximately $23 \times 23$ m$^2$. Figure 6 shows a 2D map of the study area. There are eight APs distributed in the area, as illustrated in Figure 6 with red symbols. The type of APs deployed is TP-link, and all the APs were installed at an approximate height of 2.7 m.

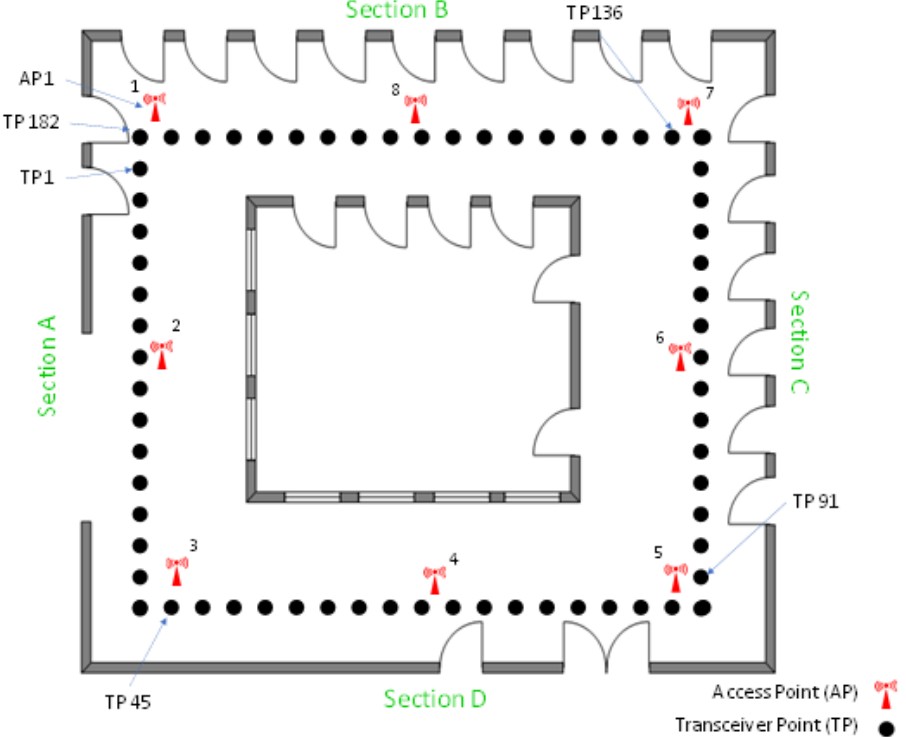

**Figure 6.** Map showing locations of the access points (APs) and transceiver points (TPs).

Track with Predetermined Checkpoints

The robot can follow a predetermined path marked by a line, with the addition of checkpoints at which the robot must stop in a specific order. The robot uses the sensors to detect the line and checkpoints and adjusts its movement based on the feedback received from the sensors. Infrared sensors are used to detect the contrast between the line and the ground. The robot is designed to follow the line by moving towards the center of the line; when a sensor crosses the line, the robot adjusts its movement to bring the robot to the center of the line. In addition to following the predetermined path, the infrared sensors are used by the controller to detect when there is a checkpoint mark on the line, where it stops and performs its desired task. A plus (+) sign along the path is used to indicate a stopping point, and an algorithm using the existing infrared sensors can detect those checkpoint marks as described in the next section.

Overall, the robot can follow a predefined path marked with checkpoints, making it useful for repetitive tasks such as the case described in this research.

The robot is set to transmit the selected WiFi-beacon parameters and the robot's position to the server. The robot surveys the floor at a speed of about one checkpoint per minute, at about 1 cm/s on average. Most of the time is spent on collecting the required samples, whereas a fraction of the time (~5%) is spent on the actual movement. The robot moved along a tracking line with a length of 82 m and TPs 0.45 m apart for this study. The main goal is to use the collected WiFi samples to train an AI model to estimate the exact location of any handheld device in 3D space using WiFi-beacon frames. The actual AI modelling, training, and validation are the subject of a forthcoming study.

### 3.3. Backend Database and Web-Server Interface

The server side was implemented using a MySQL/PHP/HTML trio, which is a standard arrangement for providing a dynamic web interface for quick storage and retrieval of all measured metrics, as demonstrated in Figure 7.

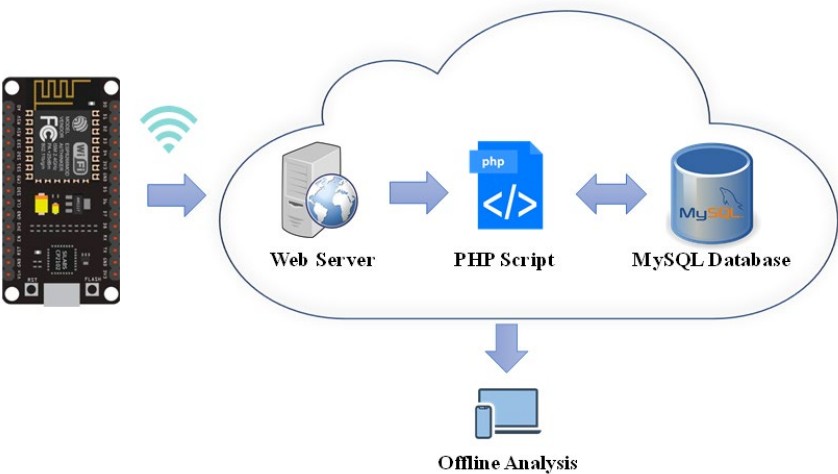

**Figure 7.** Remote web-server flow.

The backend server had two web interfaces:

1. A database-insertion interface with which the robot behaves as a client, where it can connect and submit compressed POST HTML requests that include all scanned access points for this iteration. The server in turn parses each request into tens of DB insertion commands, one for each scanned piece of access-point information.
2. A set of processing and filtering scripts through which one can quickly retrieve a selected subset of the measured metrics for further processing and analysis.

Figure 8a,c show the metadata of the main table in the database and a sample of the recorded metrics, respectively. All recorded metrics are stored in one flat table, which greatly simplifies the implementation of the PHP scripts for manipulating and retrieving information from the database. Figure 8b indicates the next "autoindex" of the main table, which indicates that over 261 K samples were collected as part of this study.

| | # | Name | Type | Collation | Attributes | Null | Default |
|---|---|---|---|---|---|---|---|
| ☐ | 1 | id 🔑 | int(11) | | | No | *None* |
| ☐ | 2 | timestamp | timestamp | | | No | *current_timestamp()* |
| ☐ | 3 | stop_number | int(11) | | | No | *None* |
| ☐ | 4 | scan_number | int(3) | | | Yes | *NULL* |
| ☐ | 5 | ssid_number | int(11) | | | No | *None* |
| ☐ | 6 | ssid_name | varchar(100) | latin1_swedish_ci | | No | *None* |
| ☐ | 7 | signal_strength | int(11) | | | No | *None* |
| ☐ | 8 | encryption | int(11) | | | No | *None* |
| ☐ | 9 | mac | varchar(100) | latin1_swedish_ci | | No | *None* |
| ☐ | 10 | height | int(4) | | | No | *None* |

(**a**)

**Row statistics**

| | |
|---|---|
| **Format** | dynamic |
| **Collation** | latin1_swedish_ci |
| **Next autoindex** | 261,269 |
| **Creation** | Nov 13, 2022 at 03:58 PM |
| **Last update** | Apr 10, 2023 at 02:12 AM |
| **Last check** | Apr 10, 2023 at 02:12 AM |

(**b**)

| id | timestamp | stop_number | scan_number | ssid_number | ssid_name | signal_strength | encryption | mac | height |
|---|---|---|---|---|---|---|---|---|---|
| 237176 | 2022-11-14 02:31:00 | 161 | 30 | 3 | TP-LINK_8F2388 | -85 | 7 | F8:D1:11:8F:23:88 | 90 |
| 237177 | 2022-11-14 02:31:00 | 161 | 30 | 4 | TP-LINK_5DC177 | -78 | 7 | 64:70:02:5D:C1:77 | 90 |
| 237178 | 2022-11-14 02:31:00 | 161 | 30 | 5 | TP-LINK_5DC1C7 | -54 | 7 | 64:70:02:5D:C1:C7 | 90 |
| 237179 | 2022-11-14 02:31:00 | 161 | 30 | 6 | TP-LINK_5DC0F5 | -59 | 7 | 64:70:02:5D:C0:F5 | 90 |
| 237180 | 2022-11-14 02:31:00 | 161 | 30 | 7 | TP-Link_D302 | -57 | 4 | 00:5F:67:0D:D3:02 | 90 |

(**c**)

**Figure 8.** (**a**) Collected metadata information; (**b**) statistics of the database's main table; (**c**) main database-table structure and few recorded metrics.

Figure 9 shows the flowchart of inserting data into the database, which is triggered by all the WiFi transceivers on the robot. This interface intended for the use of the robot is minimalistic, with simple inserts of the raw data transparently into the database.

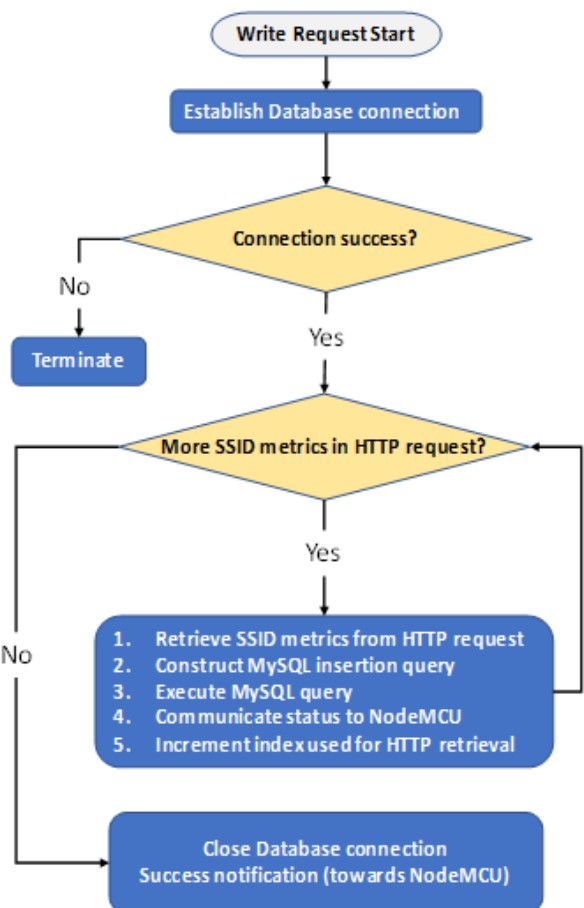

**Figure 9.** PHP flowchart intended for the use of the robot.

The link speed with the robot was about 50 Mbps, and the insertion of the data into the database did not utilize the full bandwidth. Each transmitted data packet was about 1 KB, conveying information of a full scan with all SSIDs and their signal strength. There were exactly 32 of those packets transmitted each minute, resulting in an average bit rate of less than 5 Kbps. Most of the delay was due to the nodeMCU's scanning algorithm. Furthermore, a dedicated uplink was used for transmitting the data to the server, but there may have been congestion issues in the global network that could have influenced the transmission delay and latency. However, latency, delay, timing, or congestion of the transmission/network were not taken into consideration in our analysis. We focused on reliability of the link with safeguards implemented as part of the robot's transmitting algorithm. Nevertheless, time stamps were collected for possible future use in an AI model.

Figure 10 shows the flowchart for retrieving data from the database, which is triggered by users for post-processing of the recorded signal-strength information. The retrieval process is slightly more structured towards filtering the database for a selected set of known Aps, as indicated in Figure 10. In addition, the retrieval process organizes the data based on the height of the reporting-client sub-system in the collecting robot.

All the data collected for this study, along with the firmware of the robot's main controller and client sub-systems, are now available under the ANNU-Beacon project on GitHub, as indicated earlier. This project portal will be used to publish any necessary source code and reference material that might be needed for reproducing this work.

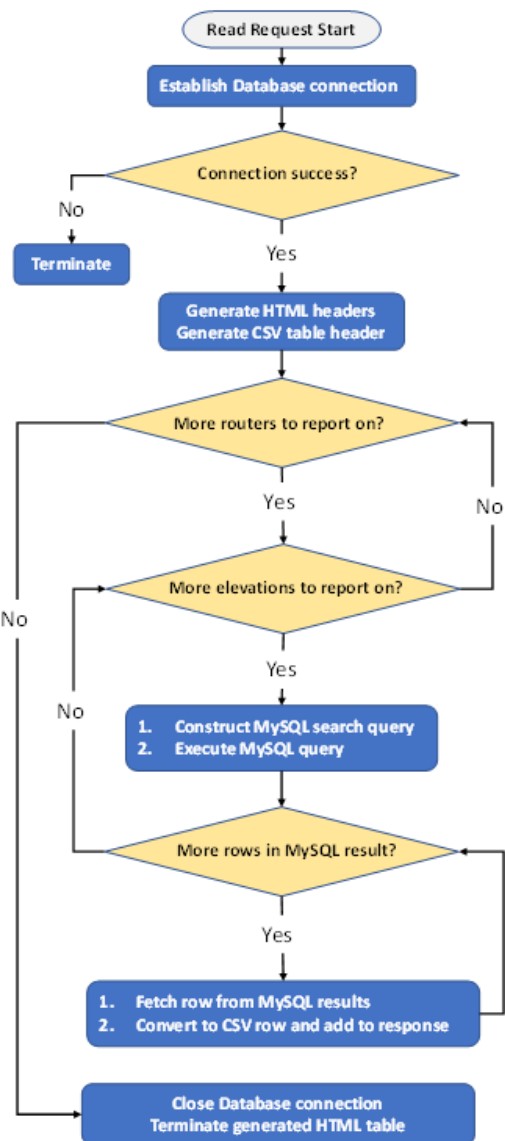

**Figure 10.** PHP flowchart for offline post-processing.

## 4. Results and Discussion

The data collection and analysis framework proposed in this paper is exceptional in terms of:

1.  The needed time, effort, and cost: the total needed time for data collection using the proposed robot-based system was approximately 3 h, which introduced a clear acceleration of the data-collection process compared to what could be collected manually. Specifically, about one minute was needed to collect 32 RSSI samples from all routers in the vicinity at three different heights, which indicates the robot's efficiency. There were about 1.5 K beacons collected, processed, transmitted to the cloud, and stored in the database. In addition, using the proposed robot was the ideal solution for such a repetitive task. The effort was minimized since the robot moved autonomously along the predefined area of study. The transceivers used were off-the-shelf low-cost NodeMCUs, which facilitate the research and development of state-of-the-art projects in developing communities, as was the case in this project. The cost of placing and measuring the track for the robot was marginal since it was guided by the layout of the existing floor tiles and was done in parallel to the robot's operation.
2.  Highlighting the effect of the transceiver height on the RSSI value: For efficient real indoor-positioning systems, the height of the APs is usually fixed, but the transceiver

height is always changing based on the handheld device, its user, and its posture (sitting, standing, young child, short/tall person, etc.). This results in different received power values. To verify the effect of different heights on the RSSI values, three different heights in a real scenario were studied in this paper: the height of the handheld device when the person is sitting at the ground, the height when a child is holding the device while standing, and the height when an adult is holding the device while standing. The corresponding heights were 0.3, 0.9, and 1.5 m, respectively. The transmitter AP's height was modelled to 2.7 m in this paper and the physical location of the APs were fixed to that height.

Figures 11–13 show the value of the received RSSI from different APs at the three heights.

It can be seen in Figures 11–13 and Tables 2–4 that the value of the RSSI varied significantly at a selected checkpoint when the transceiver height was changing. This is a very important observation that should be taken into consideration in any indoor-positioning system. The sub-figures of Figures 11–13 demonstrate a section of the captured data where the signal strength was at its minimum (the associated with strongest Wi-Fi signal). The sub-figures clearly show that the highest collection point was associated with the strongest signal. In Tables 2–4, signal strengths at selected stop points are highlighted. The stop points highlighted in the tables were spread over the whole indoor environment of the study. The tables clearly support the observation that the signal strengths increased as the height of the transceiver device increased. Therefore, we can conclude that the height of 1.5 m had the strongest RSSI value, as the distance between the AP and the receiver was the shortest.

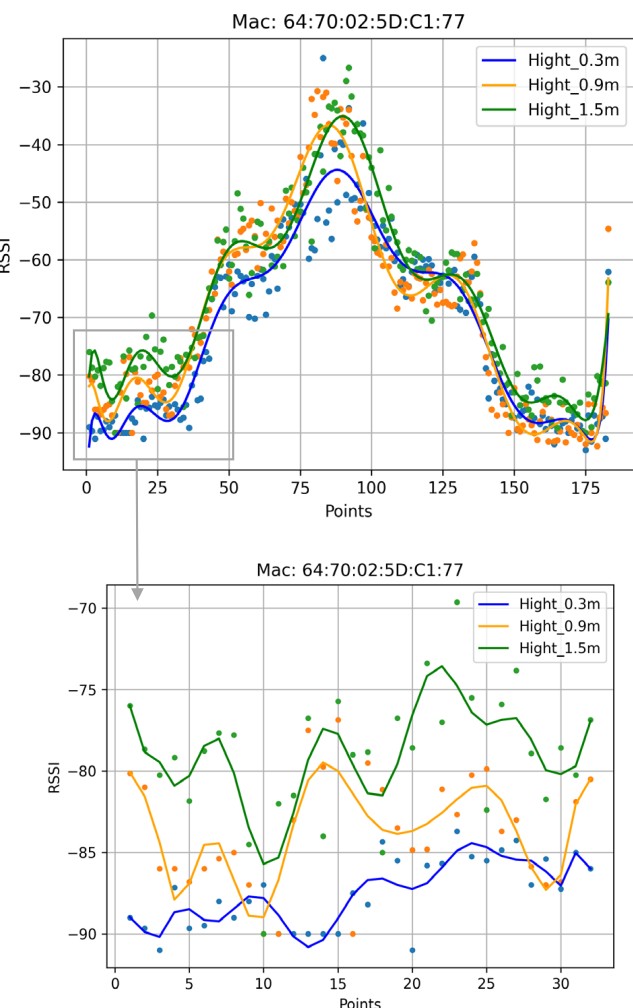

**Figure 11.** RSSI values from AP (5) at different heights.

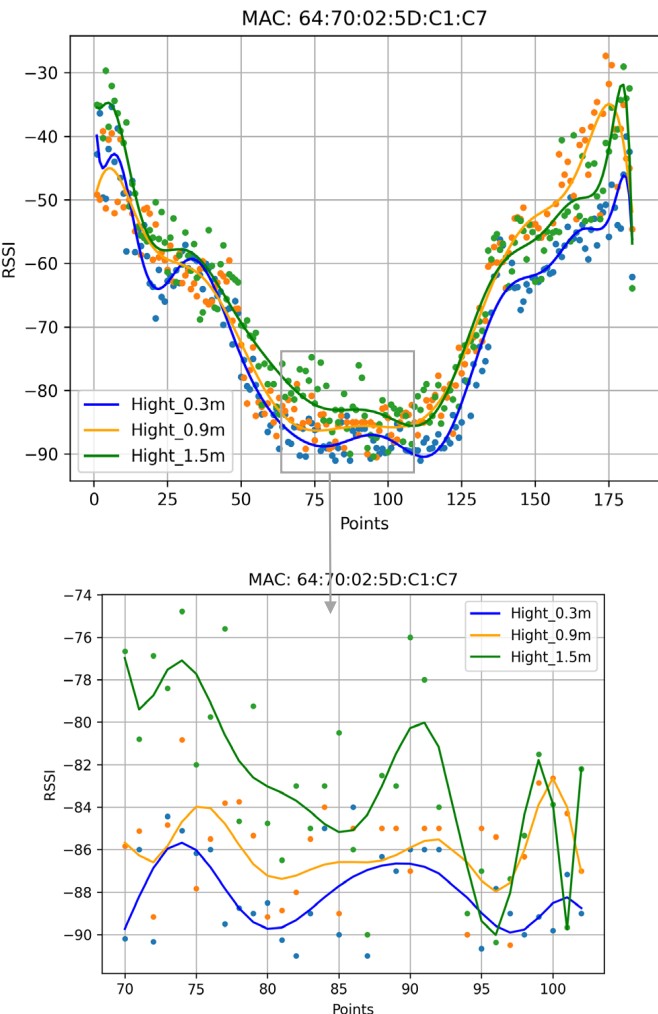

**Figure 12.** RSSI values from AP (1) with three different heights.

**Table 2.** Comparison between the three different heights at selected checkpoints from AP (5).

| Checkpoint Number | Sample Mean of RSSI, dBm at Different Heights | | |
|:---:|:---:|:---:|:---:|
| | Height 0.3 m | Height 0.9 m | Height 1.5 m |
| 1 | −89 | −80 | −76 |
| 23 | −83.7 | −82.6 | −69.6 |
| 72 | −52.7 | −50 | −47.45 |
| 91 | −48.7 | −36.35 | −29 |
| 112 | −66.4 | −64.25 | −60 |

**Table 3.** Comparison between the three different heights at different checkpoints from AP (1).

| Checkpoint Number | Sample Mean of RSSI, dBm at Different Heights | | |
|:---:|:---:|:---:|:---:|
| | Height 0.3 m | Height 0.9 m | Height 1.5 m |
| 14 | −58.17 | −54.95 | −49 |
| 44 | −69.6 | −62 | −57.13 |
| 77 | −89.5 | −83.8 | −75.6 |
| 82 | −91 | −88 | −83 |
| 98 | −90 | −86.3 | −85.3 |

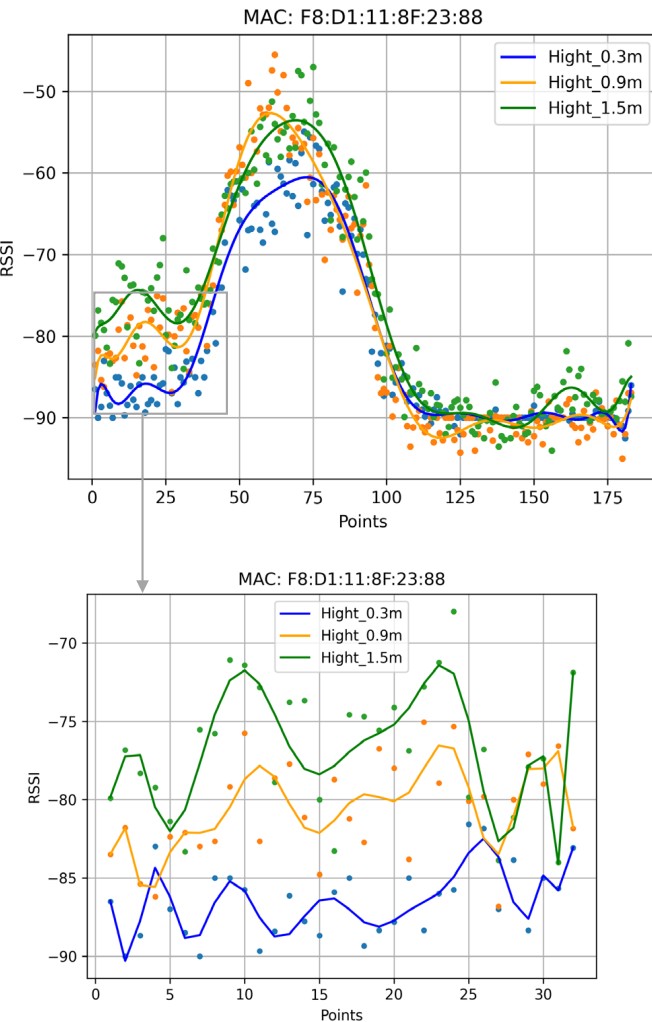

**Figure 13.** RSSI values from AP (4) at different heights.

**Table 4.** Comparison between the three different heights at different checkpoints from AP (4).

| Checkpoint Number | Sample Mean of RSSI, dBm at Different Heights | | |
|---|---|---|---|
| | **Height 0.3 m** | **Height 0.9 m** | **Height 1.5 m** |
| 1 | −86.5 | −83.5 | −79.9 |
| 5 | −87 | −82.37 | −81.4 |
| 13 | −86.12 | −77.72 | −73.8 |
| 20 | −87.8182 | −78 | −74.12 |
| 25 | −81.57 | −80.08 | −79.85 |

For all of the data, more than 65% of the checkpoints had the maximum power at the height of 1.5 m, and the remaining checkpoints had weak RSSI values, with very little difference between the three heights. Figure 14 shows the percentage of checkpoints that had the maximum RSSI value at 1.5 m.

The variation in the data values was attributed to factors in the real indoor environment in which the data were collected. Such factors are not typically found in a laboratory environment. For example, the real indoor environment had obstacles such as human movements, doors, and windows, in addition to the effect of the building material in different sections of the scanned building. In general, the shortest distance between the transmitter and receiver results in the maximum power and clearly demonstrates the effect of the height of the handheld device on the signal strength, whereas as the handheld

device moves away from the AP and approaches a new one, the signal strength of the first AP as perceived by the device is reduced in general and the effect of the height of the transceiver/handheld device becomes negligible. The use of the robot in capturing the signal strength has its strengths, since due to the systematic approach of using the robot, many sources of errors related to a human operator are eliminated. However, there may still be slight differences between the data collected by the robot compared to similar data collected from a human's handheld device due to different polarization, nature and speed of movement, and variation in height and location, amongst other factors.

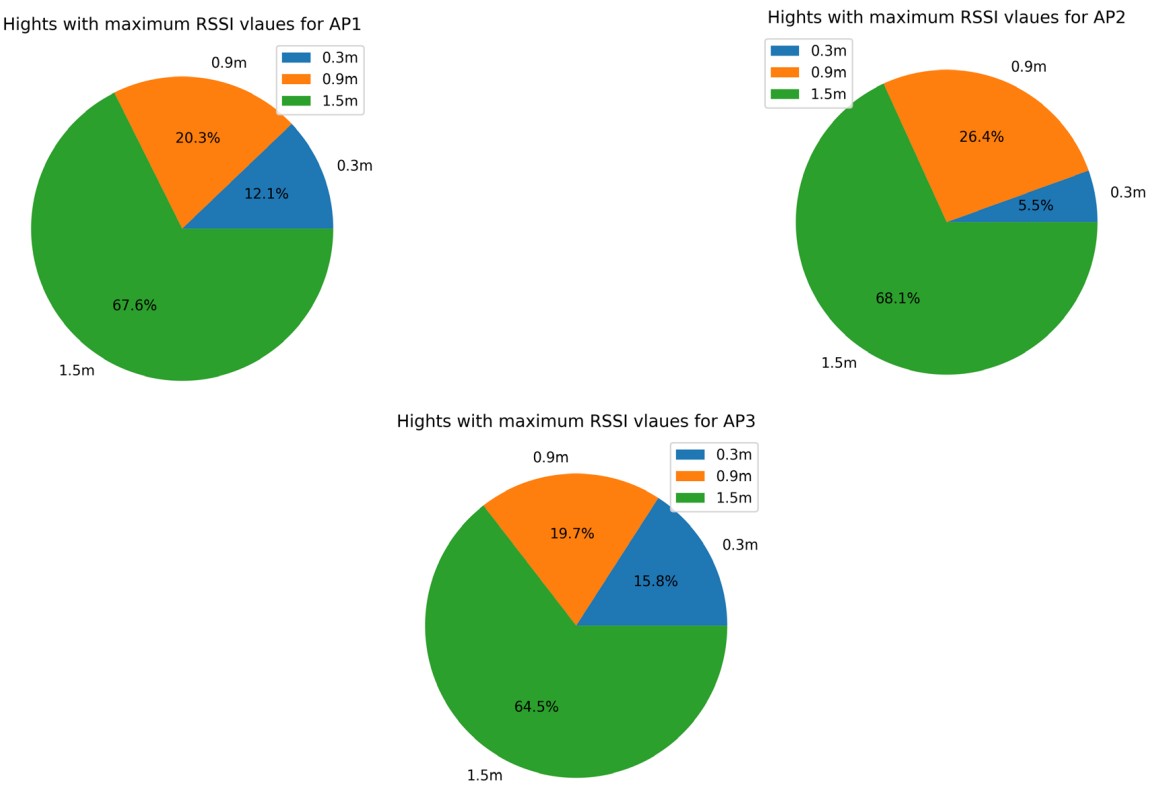

**Figure 14.** Heights with maximum RSSI values along with the checkpoint line from three different APs.

It is worth noting that the Wi-Fi infrastructure that was the subject of this study is static, with the locations of APs and their channel chosen to reduce interference and generally fixed. In addition, there is a very limited number of routers with minimum interference between them (in the range of 2–3 dBM).

## 5. Conclusions and Future Work

The work in this paper demonstrated a low-cost, scalable, and efficient framework for data collection, analysis, and potentially offline training of various indoor metrics such as, but not limited to, WiFi- and Bluetooth-beacon metrics, building geometrics, and environmental aspects.

The specific WiFi-beacon data and associated signal strength collected as part of this work will be a valuable asset for future use in an AI model for the indoor-localization process. In addition, the paper demonstrates an automated procedure for collecting many samples per point of interest at different elevations. This, in turn, lays the groundwork for an accurate 3D localization framework efficiently. This paper is an excellent proof of concept that there is a strong correlation between the receiver height and the corresponding RSSI values.

Future work includes further analysis of the collected dataset, expanding the dataset to include the whole faculty building, and improving on the robot's capabilities to navigate unassisted on any tacks.

**Author Contributions:** Conceptualization, S.A.K., E.N., M.A. and S.T.; Methodology, S.A.K., E.N., M.A. and S.T.; Software, S.A.K. and E.N.; Validation, S.A.K., E.N. and S.T.; Investigation, S.A.K., B.S. and S.T.; Resources, S.A.K.; Data curation, B.S.; Writing—original draft, S.A.K., E.N., B.S. and M.A.; Writing—review & editing, S.A.K., E.N. and M.A.; Visualization, B.S. and M.A.; Supervision, E.N.; Funding acquisition, S.A.K. and M.A. All authors have read and agreed to the published version of the manuscript.

**Funding:** This research received no external funding.

**Institutional Review Board Statement:** Not applicable.

**Informed Consent Statement:** Not applicable.

**Data Availability Statement:** Not applicable.

**Acknowledgments:** We acknowledge the support of An Najah National University.

**Conflicts of Interest:** The authors declare no conflict of interest.

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
