# Peer review of "Indoor WiFi-Beacon Dataset Construction Using Autonomous Low-Cost Robot for 3D Location Estimation"

_applsci, doi:10.3390/app13116768_

Round 1

Reviewer 1 Report

This paper proposes a framework for the autonomous construction of a diverse, extensive and open dataset with built-in redundancy. It consists of a low-cost robot with WiFi transceiver nodes for RSSI collection. The data is collected at different heights and streamed to the database. The authors have provided a very good literature review of the RF based localisation systems where robotic platforms have been used for automated measurement collection. In addition, a test setup was prepared and measurements were performed, analysed and compared in different scenarios. The topic of the paper is interesting and addresses many real problems related to indoor positioning systems and the limitations of existing solutions. However, the paper lacks novelty and the authors should address additional issues listed below to address potential reader concerns about the applicability of the proposed method.

Comments:
    1) Only the static antenna position is considered in this work, which does not fully address the problem of the real situation where the handheld device can be oriented in a different way from the stations with fixed positions. In the case of linear polarisation of the antenna, it's a crucial problem that strongly affects the localisation results due to the large differences in RSSI's collected in different positions,
    2) The authors should elaborate more on the way the RF transceivers are mounted, as the presence of the holder and the material it's made of can significantly affect the measured RSSI values,
    3) What were the Wi-Fi channels, did the authors observe any potential interference resulting in higher PER values? In the reviewer's experience this could be an issue.
    4) How much data was sent per second - what was the transmission time of the frames from the APs?
    5) It's not clear from the text how the proposed framework mitigates possible network congestion in the event of increased network traffic.
    6) Please correct the reference in lines 157 and 164.
    7) Please correct the figures and tables as the resolution is low and the style is far from that expected in a scientific publication, e.g. fonts should be aligned, the tables are just copied from the Excel sheet.
    8)It's not convincing that using the robot to follow the lines would be a huge automation of the measurements, because the reference positions have to be defined somehow, the signs/lines have to be prepared by humans, which is very time-consuming and not valuable in terms of large indoor spaces. Moreover, the imperfect placement of the signs can cause unexpected robot movements, e.g. it can change the rotation angle, which can also affect the measurement result. At least the time needed to prepare the setup for the robot should be discussed.

Reviewer 2 Report

The authors propose a framework for automated fingerprinting dataset collection, using an autonomous robot. The paper is well-written and the methodology is well-described. However, this paper has the following weak aspects:

1. The robot navigates autonomously following a marked path. Since the main claim is the efficiency of the dataset acquisition, the authors should include in their analysis the cost of placing and measuring the markers. 

2. The authors should provide a comparative analysis with other methods, e.g., human-based data collection.

3. The authors should extend with a recent literature review.

4. Errors with not found references.

None.

Reviewer 3 Report

No algorithm innovation, only experimental analysis

Reviewer 4 Report

Authors focus their work on how the strength of the received WiFi signal is affected by the height of the handheld WiFi antenna, which is connected to a robot, where different sets of antenna heights are considered. Authors have created a framework for au tonomous construction of a diverse, extensive, and open dataset.

It is an interesting research although there are some issues that must be fixed:

Authors should include a paragraph with the explanatino of the structure of the paper at the end of the introduction section.

I finnd missing thiis paper in the related work section:

Support vector regression for mobile target localization in indoor environments, Sensors 22 (1), 358. 2022

In section 3 there are several errors to references: "Error! Reference source not found.."

Figure 1 is not cited from the text.

Authors should discuss the errors of the measurements.

After Figure 11, the number of figure 12 seems to be missing.

Figures 11, 12, and 13 must be discussed in detail. It also happens with table 1, 2 and 3.

Authors should include their future work at the endo of the conclusion section.

The paper is easy to read. I have not found mistakes.

Round 2

Reviewer 1 Report

The authors have addressed most of the reviewer comments. I'm still not convinced about considering the proposed approach as significantly more efficient when comparing to what has already been described in the literature. However, I can accept the explanation that the robotic platform or the way of automated measurements can be easilly changed in the future without modification of the whole setup. Please check and correct the formatting from line 505.

Reviewer 4 Report

Authors have fixed all my comments. The paper is ready to be published.

I do not find mistakes in the English Language
